# Genetic Approach to Elucidate the Role of Cyclophilin D in Traumatic Brain Injury Pathology

**DOI:** 10.3390/cells10020199

**Published:** 2021-01-20

**Authors:** Ryan D. Readnower, William Brad Hubbard, Olivia J. Kalimon, James W. Geddes, Patrick G. Sullivan

**Affiliations:** 1Spinal Cord and Brain Injury Research Center (SCoBIRC), University of Kentucky, Lexington, KY 40536, USA; ryan.readnower@gmail.com (R.D.R.); bradhubbard@uky.edu (W.B.H.); olivia.kalimon@uky.edu (O.J.K.); jgeddes@uky.edu (J.W.G.); 2Department of Neuroscience, University of Kentucky, Lexington, KY 40508, USA; 3Lexington Veterans’ Affairs Healthcare System, Lexington, KY 40502, USA

**Keywords:** controlled cortical impact, mitochondrial bioenergetics, mitochondria, cyclosporin a, NIM811, mitochondrial permeability transition, Ppif, neuroprotection

## Abstract

Cyclophilin D (CypD) has been shown to play a critical role in mitochondrial permeability transition pore (mPTP) opening and the subsequent cell death cascade. Studies consistently demonstrate that mitochondrial dysfunction, including mitochondrial calcium overload and mPTP opening, is essential to the pathobiology of cell death after a traumatic brain injury (TBI). CypD inhibitors, such as cyclosporin A (CsA) or NIM811, administered following TBI, are neuroprotective and quell neurological deficits. However, some pharmacological inhibitors of CypD have multiple biological targets and, as such, do not directly implicate a role for CypD in arbitrating cell death after TBI. Here, we reviewed the current understanding of the role CypD plays in TBI pathobiology. Further, we directly assessed the role of CypD in mediating cell death following TBI by utilizing mice lacking the CypD encoding gene *Ppif*. Following controlled cortical impact (CCI), the genetic knockout of CypD protected acute mitochondrial bioenergetics at 6 h post-injury and reduced subacute cortical tissue and hippocampal cell loss at 18 d post-injury. The administration of CsA following experimental TBI in *Ppif*-/- mice improved cortical tissue sparing, highlighting the multiple cellular targets of CsA in the mitigation of TBI pathology. The loss of CypD appeared to desensitize the mitochondrial response to calcium burden induced by TBI; this maintenance of mitochondrial function underlies the observed neuroprotective effect of the CypD knockout. These studies highlight the importance of maintaining mitochondrial homeostasis after injury and validate CypD as a therapeutic target for TBI. Further, these results solidify the beneficial effects of CsA treatment following TBI.

## 1. Introduction

### 1.1. Secondary Mitochondrial Cascades in Traumatic Brain Injury

Traumatic brain injury (TBI) is a major health concern that affects significant numbers of people worldwide. According to the Centers for Disease Control and Prevention (CDC), in the United States alone, there are an estimated 3 million TBI-related emergency department visits, hospitalizations, and deaths yearly [1]. There are no current approved treatments for TBI due to the complexity of the secondary injury cascade following primary head injury. One important player in this injury cascade is the mitochondrion. Known as the “powerhouse” of the cell, mitochondria are critical in regulating cellular energy homeostasis, redox balance, calcium buffering, and cell death [2]. In the secondary phase of brain injury, there is a bioenergetic collapse resulting from disrupted intracellular calcium homeostasis and increases in oxidative stress. Mitochondria sequester increased levels of intracellular calcium, concomitant with excitotoxic mechanisms after TBI. Once the mitochondria can no longer take up any more calcium, they become dysfunctional, generate ROS, and induce mitochondrial permeability transition (mPT).

### 1.2. Role of mPT in TBI Pathophysiology

The mitochondrial permeability transition pore (mPTP) is a non-selective channel in the inner mitochondrial membrane, which leads to mitochondrial swelling and rupture of the outer mitochondrial membrane. In turn, an outer membrane rupture allows the release of calcium, cytochrome c, and other solutes (<1.5 kDa) from the matrix into the cytosol. This results in mitochondrial swelling, dysfunction, rupture, and eventually cell death [3]. The release of cytochrome c through the pore into the cytosol activates downstream caspases, which also triggers apoptotic events, adding to CNS injury pathology, further implicates mPT as an important target for neuroprotection after TBI [4,5]. Since mitochondria are arbitrators of both apoptotic and necrotic cell death, they have become targets for therapeutic intervention in TBI and other neurodegenerative diseases [6]. Under stress conditions (such as high mitochondrial calcium or increased reactive oxygen species (ROS)), cyclophilin D (CypD) purportedly interacts with the adenine nucleotide translocase (ANT) and/or the F-ATP synthase of the inner mitochondrial membrane. The interaction with ANT results in a conformational change converts ANT into a non-specific pore located in the inner membrane. Following the voltage dependent anion channel (VDAC) interacts with the CypD/ANT complex, which promotes formation of the mPTP [7,8]. Other groups have confirmed the ability of ANT to form a large ion channel and constitute (or at least contribute) to mPT [9,10]. However, recent evidence suggests that while these mitochondrial inner membrane proteins contribute to mitochondrial desensitization to high calcium levels, they may not constitute the primary mPTP [11,12]. After calcium overload, CypD can bind to the oligomycin sensitivity-conferring protein (OSCP) subunit of the ATP synthase and this interaction has been suggested to trigger the opening of a large conductance channel found in ATP synthase, which is potentially a major component of the mPTP [13,14,15]. Conversely, multiple groups have shown that mPTP can form in the absence of key subunits of the F-ATP synthase [16,17,18] and the assembled ATP synthase itself [19]; thus, the molecular identity of the mPTP remains highly disputed. Nevertheless, induction of mPT results in a loss of the mitochondrial membrane potential, resulting in the uncoupling of electron transport from ATP production. Moreover, mPT leads to mitochondrial swelling, rupture of the outer mitochondrial membrane, release of pro-apoptotic molecules (i.e., cytochrome c), and increased ROS production. Early studies found that CsA interacts and binds to CypD to inhibit mPT [20,21]; therefore, CypD has emerged as a therapeutic target in TBI.

### 1.3. Therapeutic Targeting of mPT by Cyclosporin A

CypD, a target of the FDA approved immunosuppressant cyclosporin A (CsA), has been shown to play a key role in the modulation of mPTP formation [7,22,23]. CypD belongs to a family of proteins known as peptidyl-prolyl cis-trans isomerases (PPIases) and is localized in the mitochondrial matrix. This was elucidated by the observed effect of mPTP desensitization after CsA administration, requiring higher levels of intra-mitochondrial calcium to initiate pore formation [24]. The immunosuppressive properties of CsA were not shown to be the cause of neuroprotection after another study was performed with a more potent immunosuppressor, FK506. Both CsA and FK506 suppress T cell activation, but only CsA offered neuroprotection, most likely due to action on the mPTP [25]. There is an abundance of preclinical evidence that demonstrates that mPT inhibition following TBI is neuroprotective. Administration of CsA to inhibit mPT following TBI has proven to be effective at improving mitochondrial function and neuronal survival in multiple models of TBI [25,26,27,28,29,30,31,32,33,34,35]; however, there are contradictory results on the cognitive effects following TBI and CsA administration [27,36]. The neuroprotective effects of CsA are limited due to toxicity at high doses [37,38] but Phase II trials involving 5 mg/kg of CsA have shown efficacy in TBI patients administered within 8 h post-injury [39]. Alternatively, in another Phase II study in severe TBI patients, administration of CsA after 12 h did not show any significant improvements in neurological outcomes [40]. In a rat model of spinal cord injury (SCI), the optimal dose and regimen of CsA determined by Sullivan et al., did not offer any beneficial effects in tissue sparing, which contradicted results obtained following TBI [33,41]. Among further investigation, it was found that isolated brain and spinal cord mitochondria are different; more specifically, in lipid peroxidation, mitochondrial mRNA count, complex-I activity, and calcium sequestration [42]. Spinal cord mitochondria were shown to form the mPTP at lower concentrations of calcium, with the addition of CsA offering only slight inhibition at doses used in TBI animal models. This is in stark contrast to cortical mitochondria, which require more calcium to form the pore and is significantly inhibited by CsA. One possible explanation for this difference is that there is more CypD mRNA in the spinal cord than in the cortex. In order for CsA to show neuroprotective effects in the SCI model, the optimal dose should be increased; however, due to its toxicity, this may not be a feasible therapeutic option [43]. Additionally, CsA has been shown to bind to other targets such as the T-cell activator, calcineurin, which makes pinpointing the mechanism(s) of action of CsA related to neuroprotection challenging [44].

### 1.4. Therapeutic Targeting of mPT by NIM811

A non-immunosuppressive analog of CsA, *N*-methyl-4-isoleucine-cyclosporin (NIM811), has also been utilized to observe the effects of CypD inhibition on TBI outcomes. NIM811 is also tolerated better at high concentrations compared to CsA, most likely due to the substitution of isoleucine in place of leucine; this alternate prevents NIM811 from binding to calcineurin, the enzyme conferring CsA’s immunosuppressive properties [45]. However, NIM811 binds CypD as well as CypA, another member of the cyclophilin, peptidylprolyl isomerase (PPI) family [46]. NIM811 improved motor function, improved mitochondrial function, decreased oxidative damage, and decreased neurodegeneration following TBI [47,48]. Results from these studies have shown NIM811 to have similar effects as CsA on preserving mitochondrial function after TBI. Our group has also determined that administration of a single dose of 10 mg/kg NIM811 is sufficient for improving mitochondrial respiration after experimental TBI [46]. In addition, dosing 15 min and 24 h after injury improved cortical tissue sparing and performance in the Morris water maze (MWM) test, implying translational efficacy of this compound after TBI [46]. NIM811 also improved outcome measures following SCI in rats albeit at higher doses [49,50,51]. Presumably, the neuroprotective actions of CsA and NIM811 are attributable to their inhibition of CypD, which prevents the binding of CypD to ANT and formation of the mPTP [52]. However, it is difficult to discern the protective mechanisms of NIM811 and CsA given that both have targets other than CypD. Further, as cell death may occur independent of CypD, direct evidence regarding the role of CypD in mPTP formation in the context of TBI is needed.

### 1.5. The Effects of CypD Knockout (KO) in Neurodegenerative Diseases

In order to examine the direct role of CypD in mPTP formation, CypD knockout mice lacking the encoding gene, *Ppif*, have been used by researchers. Studies have demonstrated the role CypD plays in mPTP formation and the subsequent effects on calcium uptake of mitochondria [23,53]. Interestingly, CypD-null mitochondria were not completely resistant to mPTP formation even though they were desensitized to calcium stress. Compared to brain mitochondria from WT mice, mitochondria from CypD KO mice retained 30–40% more calcium when given in 10 µM boluses. Additionally, CypD KO mitochondria required 80 µM of calcium to diminish the membrane potential, whereas the wild-type mice only required 50 µM calcium. Further, CsA treatment did not increase the calcium threshold in CypD KO mitochondria nor their ability to maintain membrane potential [54]. CypD knockout mice were indeed insensitive to CsA, providing further evidence of its action on CypD [53]. Studies in CypD knockout mice have demonstrated that mitochondria lacking CypD are resistant to Ca^2+^- and ROS-induced mPT [23,53,54]. In corroboration, the CypD knockout increased mitochondrial resistance to mPTP opening during cardiac ischemia/reperfusion (I/R) injury and lowers necrosis [22]. While the CypD knockout significantly reduces infarct area after cardiac I/R injury, which is predominantly driven by Ca^2+^ and oxidative stress, it did not protect against cell death induced by pro-apoptotic Bcl-2 family members [23]. CypD-mediated sensitization to mitochondrial calcium stress can also be observed in brain mitochondria, where the CypD knockout improved mitochondrial membrane polarization and survival due to desensitization to increased calcium levels [54,55]. This indicated that cell death may occur through either CypD-dependent or CypD-independent mechanisms but reinforced the close-knit relationship between CypD and mPTP opening. 

To elucidate the role CypD plays in neuropathology following TBI, we subjected mice lacking the CypD encoding gene *Ppif* to controlled cortical impact (CCI). We hypothesize that genetic knockout of CypD would confer neuroprotection following TBI due to inhibition of mPT. Indeed, the CypD knockout restored acute mitochondrial function following TBI. In the subacute phase following injury, cortical tissue sparing and CA3 neuron density are improved in mice lacking CypD, although this does not benefit cognitive function. The data demonstrate that knockout of CypD, likely via maintenance of mitochondrial homeostasis, spares cortical tissue and protects CA3 neurons following TBI, which supports CypD as an important mediator of cell death following TBI.

## 2. Materials and Methods

### 2.1. Animals and Experimental Design

All of the studies performed were approved by the University of Kentucky Institutional Animal Care and Use Committee (IACUC). Additionally, the Division of Laboratory Animal Resources at the University is accredited by the Association for the Assessment and Accreditation for Laboratory Animal Care, International (AAALAC, International); all experiments were performed within its guidelines. All animal experiments complied with ARRIVE (Animal Research: Reporting of In Vivo Experiments) guidelines and experiments were carried out in accordance with the National Institutes of Health Guide for the Care and Use of Laboratory Animals (NIH Publications No. 8023, 8th edition, revised 2011). *Ppif*-null (CypD knockout (KO)) mice and control C57BL/6 wild-type mice were bred at the University of Kentucky (Lexington, KY, USA), and were originally obtained as a gift from Dr. J.D. Molkentin (Cincinnati Children’s Research Foundation). Adult (~8 to 10 weeks old) male wild-type (WT) and CypD KO mice were utilized and randomly assigned to experimental groups. The animals were housed 5 (mice) per cage and maintained in a 14-h light/10 h dark cycle. All animals were fed a balanced diet ad libitum and water was reverse osmosis generated. One cohort was euthanized at 6 h post-injury for mitochondrial assessments (n = 5–6/group) while a second cohort was euthanized at 18 d post-injury, following a week of rest and subsequent behavioral analysis, for tissue sparing and hippocampal cell count analysis (n = 5/group). The final (third) cohort was administered treatment targeting mPT and was euthanized for tissue sparing assessment at 7 d post-injury (n = 7–8/group). For mitochondrial bioenergetics assessment, experiments were conducted with n = 5–6 biological replicates and for each biological replicate there were technical replicates of n > 3. All data analysis was performed blinded to treatment groups.

### 2.2. Controlled Cortical Impact

Mice were subjected to a severe (1.0 mm) unilateral controlled cortical contusion TBI or sham-operation according to past studies [56,57,58]. Briefly, mice were put under anesthesia with 2–5% isoflurane and the skull was exposed through a midline incision. An approximately 3 mm craniotomy was made lateral to the midline and centered between bregma and lambda, without disrupting the dura. A cortical contusion was produced using a pneumatically driven injury device (TBI-0310 Impactor, Precision Systems and Instrumentation (PSI), Fairfax, VA USA) with a 2 mm tip as previously described [58]. Following injury, a prosthetic skull cap was glued over the craniotomy site and incisions were sutured together. Animals remained on a 37° heating pad until they were mobile and fully responsive.

For the third cohort, mice subjected to a severe CCI were then administered vehicle (100% DMSO), NIM811 (10 mg/kg), or CsA (20 mg/kg) 15 min after TBI with a subsequent injection at 24 h post-injury.

### 2.3. Mitochondrial Isolation and Respirometry Analysis

At 6 h post-injury, the first cohort of wild-type and CypD knockout mice were asphyxiated with CO_2_ until unconscious, decapitated, and the brains were rapidly removed and placed in isolation buffer (215 mM mannitol, 75 mM sucrose, 0.1% BSA, 20 mM HEPES, and 1 mM EGTA; pH 7.2). The ipsilateral cortex was dissected with a 3 mm-diameter punch centered on the site of impact. The cortical tissue punch contained tissue from the site of the impact and the surrounding penumbra. The tissue punches were homogenized and isolated by differential centrifugation as previously described [56,59,60]. Briefly, the homogenate was centrifuged at 1300× *g* for 3 min. Following the first spin, the supernatant was placed in a fresh tube and the pellet was resuspended in isolation buffer and centrifuged at 1300× *g* for 3 min. The supernatant from the first and second spins were collected in separate tubes and spun at 13,000× *g* for 10 min. The pellets from both tubes were combined, resuspended in 400 µL isolation buffer, and placed in a nitrogen bomb at 1200 psi for 10 min. The pressure in the nitrogen bomb was rapidly released and the sample was placed as the top layer on a Ficoll separation column, which consisted of a 10% Ficoll layer and a 7.5% Ficoll layer. The Ficoll column with sample was centrifuged at 100,000× *g* for 30 min at 4 °C using a Beckman SW 55Ti rotor and ultra-centrifuge.

The supernatants were carefully removed, and the pellet was resuspended in isolation buffer without EGTA and centrifuged at 13,000× *g* for 10 min at 4 °C. In order to completely wash all Ficoll out of the sample, the mitochondrial pellets were recentrifuged at 10,000× *g* for 5 min at 4 °C. The final mitochondrial pellet was resuspended in isolation buffer without EGTA to yield a concentration of ~10 mg/mL. The protein concentration was determined using a bicinchoninic acid protein assay kit (Pierce, Rockford, IL, USA).

Mitochondrial respiration was measured using a Seahorse Biosciences XFe24 Flux Analyzer (North Billerica, MA, USA) as previously described [46,56,60]. Briefly, 5 µg of mitochondrial protein were added to each well in 50 µL respiration buffer. The assay plates were centrifuged at 3000 rpm for 4 min at 4 °C. Additional respiration buffer was added to bring the starting volume to 475 µL. After calibration, the assay plate was placed and pyruvate/malate/ADP, oligomycin, FCCP, and rotenone/succinate were injected sequentially through ports A–D, respectively. The final concentrations of substrates and inhibitors were 5 mM (pyruvate), 2.5 mM (malate), 1 mM (ADP), 1 μg/mL (oligomycin), 1 μM (FCCP), 100 nM (rotenone), and 10 mM (succinate). Oxygen consumption rates (OCR) were recorded in each distinct respiration state.

### 2.4. Morris Water Maze

A variant of the MWM task was used to assess cognitive function/dysfunction following TBI in these experiments [61]. The maze was in low light and consisted of a circular pool filled with water (27 °C). A platform was placed below the water surface rendering it invisible and was used as the goal platform. The pool was situated in a room that had numerous extra-maze cues that remained constant throughout the experiment. A video camera system placed directly above the center of the pool recorded swimming performance and each video record was processed by a video motion analyzer (Ethovision-XT, Noldus, Leesburg, VA, USA). Water maze testing began 10 d after surgery and training consisted of four daily trials one each starting from a different labeled quadrant. Each trial was initiated by placing the mouse into the water in a quadrant either adjacent or opposite to the platform. The platform location was fixed throughout the training. Each trial lasted 60 s or until the mouse located the platform. Mice that did not find the platform were guided to it and given a latency score of 60 s. Each mouse was required to spend 15 s on the platform at the end of each trial. During this learning period mice used external visual cues as a reference to find the submerged platform. The latency to find the platform was recorded for each trial. On the fifth day of the MWM task, mice were given one probe trial, in which the platform was removed. A first dependent measure was the time spent swimming in each quadrant. The second dependent measure was the number of times the mouse crossed over the platform arena.

### 2.5. Tissue Processing and Tissue Sparing Measurement

At 18 days post-injury, wild-type and CypD knockout mice were anesthetized by an overdose of pentobarbital (95 mg/kg body weight) and transcardially perfused with physiological saline followed by 4% paraformaldehyde. After removal, the brains were placed in 4% paraformaldehyde-15% sucrose for an additional 24 h. Coronal sections (30 μm) were then cut using a freezing microtome, throughout the rostrocaudal extent of the brain, extending through the septal area to the most posterior extent of the hippocampus. A series of coronal tissue sections spaced ~400 μm apart (minimum of 12 slices) were mounted on slides, stained with cresyl violet, and subjected to image analysis for lesion volume assessment. Quantitative assessment of cortical damage employed a blinded unbiased tracing protocol utilizing the Cavalieri method of segmentation to compare ipsilateral cortex to contralateral cortex. All slides were assessed blindly with respect to treatment group, for ROI analysis to measure cortical sparing, using ImageJ software (NIH, Bethesda, MD, USA).

### 2.6. Stereology—Hippocampal Cell Counts

The same series of cresyl violet-stained coronal brain slices were used for hippocampal cell counting. All sampling was conducted using an Olympus BX51 microscope with a 60 X oil objective, with an ASI automated stage (Eugene, OR, USA). The neuronal cells were distinguished from other cells based on the cell size, morphology and granular staining [62]. Bioquant Image analysis software (R and M Biometrics, Memphis, TN, USA) was used to estimate the total cell number in ipsilateral hippocampal regions CA1, CA3, and DG using the optical fractionator method as previously described [63]. Briefly, this method involves sampling a known fraction of the section thickness, under a known fraction of the sectional area, in a known fraction of the sections that contain the structure. The total number of neurons (N) is estimated by: *N* = Σ*Q* * *t/h* * 1/asf * 1/ssf, where Σ*Q* is the number of neurons counted in the optical dissectors, *t* is the tissue thickness (20 μm), h is the height of the dissector (20 μm), 1/asf is the counting grid area (100 × 100)/the dissector area (CA3, CA 1–15 μm × 15 μm; DG-10 μm × 10 μm), and 1/ssf is the sampling section fraction (12).

### 2.7. Statistics

Power analysis was conducted (using G*Power statistical software; version 3.0.10) for all experimental data. Analysis was completed based on the ANOVA or t-test. A priori analysis was performed and effect size was calculated based on expected mean ± SD within each group. Power analysis was calculated for behavioral experiments using the following parameters: α = 0.05, 1 − β = 0.8, and standard deviation 20% of mean (effect size = 1.12) for experimental groups. For all statistical comparisons, significance was set at *p* < 0.05. For each measure, data were measured using interval/ratio scales. The Brown-Forsythe and/or Bartlett’s tests were performed to ensure homogeneity of variance. Furthermore, the Shapiro-Wilk test was completed to ensure normality. As these criteria were met for all experimental data, parametric statistics were employed for all analyses. Tissue sparing assessment with WT compared to CypD KO were analyzed using unpaired *t*-test. All other data were analyzed using one-way ANOVA followed by Tukey’s post-hoc analysis.

## 3. Results

### 3.1. Cyclophilin D Knockout Improves Mitochondrial Bioenergetics

We found that brain mitochondria derived from the CypD knockout buffered higher calcium levels compared to mitochondria from WT mice (data not shown), confirming previous research. To determine the effects of the CypD knockout on mitochondrial respiration following TBI, mitochondria were isolated at 6 h post-injury from the ipsilateral cortex of wild-type and CypD knockout mice. In the presence of pyruvate/malate/ADP, injury-induced impairments (F_(2,14)_ = 8.37, *p* = 0.003 WT Sham vs. WT Injured) in complex I driven State III respiration (dependent on the rate of ATP synthase) were attenuated by the CypD knockout (F_(2,14)_ = 8.37, *p* = 0.047 WT Injured vs. CypD KO Injured) (Figure 1). There were no differences in State IV respiration (dependent upon proton leak across the inner membrane) among treatment groups. While complex II driven State V respiration (maximal rate of the electron transport chain) was significantly reduced in both WT Injured and CypD KO Injured mice (F_(2,14)_ = 9.69, *p* < 0.03), complex I driven State V respiration was only significantly reduced in WT Injured (F_(2,14)_ = 5.08, *p* < 0.03 WT Sham vs. WT Injured). The respiratory control ratio (RCR; a metric for the coupling of electron transport with ATP production) did not differ among groups, indicating a similar level of mitochondrial coupling between the groups (data not shown).

### 3.2. Cyclophilin D Knockout Alleviates TBI-Related Reduction of CA3 Hippocampal Neurons

To determine the effects of CypD knockout on hippocampal cell survival after TBI, neurons were counted for regions dentate gyrus (DG), CA3, and CA1 using the optical dissector method. The number of surviving CA1, CA3, and DG neurons were estimated at 18 days post-injury (Figure 2). In WT mice, injury significantly decreased the number of neurons in ipsilateral CA1 (F_(3,16)_ = 40.59; *p* < 0.0001), CA3 (F_(3,16)_ = 9.31; *p* = 0.003), and DG (F_(3,16)_ = 37.02; *p* < 0.0001) compared to WT Sham. In CypD KO mice, injury significantly decreased the number of neurons in ipsilateral CA1 (F_(3,16)_ = 40.59; *p* < 0.0001) and DG (F_(3,16)_ = 37.02; *p* < 0.0001) compared to CypD KO Sham. In ipsilateral CA3 of CypD KO Injured mice, neuronal count was not significantly different than CypD KO Sham mice, highlighting mitigation of the TBI-associated CA3 neuronal loss.

### 3.3. Cyclophilin D knockout Increases Tissue Sparing Following TBI

In order to determine the effects of CypD knockout on tissue sparing following TBI, cortical tissue sparing was measured at 18 days following severe CCI utilizing the Cavalieri method. CypD KO Injured mice had significantly higher tissue sparing levels (84.1 ± 2.1) compared to WT Injured mice (76.2 ± 2.248) (*t* = 2.57, *p* < 0.03) (Figure 3).

### 3.4. Cyclophilin D Knockout Does Not Improve Cognition Following TBI

To examine the effect of CypD KO on cognitive function after TBI, mice performed the MWM task on five consecutive days starting at 10 days post-injury. In general, CypD KO mice (both Sham and Injured) displayed worse performance, higher latency to find the platform, during the learning phase (Figure 4A). This is highlighted by a significantly increased average latency over the four training days in the CypD KO Injured group compared to WT Injured animals (Figure 4B; F_(3,15)_ = 7.23, *p* = 0.033). During the probe trial, there was no difference in time in the target quadrant or platform quadrant crossings amongst experimental groups (Figure 4C,D).

### 3.5. Cyclosporin A Provides Neuroprotection in CypD KO Mice after TBI

In order to determine the effects CsA and NIM811 administration in CypD KO mice after TBI, cortical tissue sparing was measured at 7 days following severe CCI utilizing the Cavalieri method. NIM811 administration after CCI in CypD KO mice did not improve tissue sparing compare to Vehicle administration (Figure 5). However, CsA treatment resulted in increased cortical sparing compared to vehicle-treated CypD KO mice (F_(2,20)_ = 3.40, *p* = 0.039).

## 4. Discussion

The results of this study uncovered the role of CypD in the pathophysiology of TBI, in which CypD-dependent cell death mechanisms have been implicated due to intracellular Ca^2+^ dyshomeostasis and subsequent mitochondrial calcium sequestration. We presented data that demonstrates that genetic ablation of CypD is neuroprotective following TBI. Further, CypD knockout attenuated injury-induced mitochondrial dysfunction. The data confirms the key role CypD plays in cell death following TBI.

Pharmacological studies have suggested an important role for CypD in neurodegeneration and TBI. Administration of CsA has been shown to be neuroprotective following multiple experimental models of TBI [25,28,29,31,33,34]. Since CsA has multiple biological targets, interacting with CypD and inhibiting calcineurin, the neuroprotective mechanism(s) of CsA is greatly debated. Indeed, we found that CsA provides neuroprotection via pathways other than CypD inhibition (Figure 5). NIM811, which inhibits CypD but does not affect calcineurin, has also been shown to have neuroprotective properties following TBI [46,47]. However, NIM811 has a U-dose response curve following TBI, which may indicate that NIM811 is acting through other potential off-target mechanisms, such as CypA. Genetic knockout of CypD provides a novel approach for directly examining the role that CypD plays without the confounding off-target effects of pharmacological studies.

The observed improvement in mitochondrial complex I activity following TBI in CypD knockout mice likely reflects a secondary effect of inhibition of mPTP formation via CypD ablation [48]. Improvement of mitochondrial complex I function suggests that loss of CypD increases mitochondrial preservation after TBI. These findings are in line with studies involving mitochondria isolated from CypD deficient mice, which showed that mitochondria lacking CypD are resistant to Ca^2+^ induced mPTP formation [23].

We found that the CypD knockout was neuroprotective at 18 days following TBI (Figure 2 and Figure 3). While previous studies examined neuroprotection at 7 days post-injury [33,35], we chose to examine neuroprotection at a later time point in order to incorporate cognitive assessment. The selective mitigation of CA3 neuronal loss observed in CypD knockout mice could be, in part, explained by the relative vulnerability of hippocampal subfields following brain contusion. Selective vulnerability of CA1 neurons is well-documented following hypoxia-ischemia insult [64,65]. In fact, it has been shown that inhibition of complex II with malonate results in a selective loss of CA1 neurons in rats [66]. However, another study showcases that CA3 is more vulnerable after CCI impact compared to CA1 [67]. Our group showed that mild mitochondrial uncoupling treatment lessens CA3 neuronal loss following severe CCI [56]. This suggests that CA3 neuronal loss may be selectively mediated through mitochondrial-dependent mechanisms and therefore more amenable to mitochondrial-directed therapy. CypD knockout reduced some, but not all, subtypes of axonal injury following mild TBI, again highlighting regional selectivity [68]. Further, the effects in CA3 may be partially explained by the predominant expression of CypD in GABAergic interneurons, including those of the CA3 [69]. It may also be pertinent to examine other brain regions in future studies as a report has demonstrated that CypD immunoreactivity is higher in the substantia nigra as compared to other regions such as cortex [69].

The CA3 region has been shown to play a critical role in spatial memory acquisition and in the formation of long-term spatial memory [70]. Since CypD knockout resulted in alleviation of CA3 neuronal loss, this suggests that modulation of CypD may improve memory following TBI. To directly address this, we performed cognitive studies using the MWM in CypD KO mice subjected to TBI. We did not observe any injury-induced impairment in cognitive function in WT animals using the MWM. Interestingly, we did observe a genotype-dependent impairment in MWM function in CypD KO mice compared to WT mice (Figure 4). Similar results were reported by Mouri and colleagues [71]. In these studies, genetic knockout of CypD resulted in cognitive impairments in several behavioral tasks. These data provide a potential physiological role for CypD, likely related to calcium signaling, and offer a mechanism by which memory is at least partially dependent on CypD function.

Although it is generally accepted that mPT plays a role in cell death, the mechanisms underlying induction of mPT following TBI remain unclear. There appears to be two alternative routes to cell death, one that is CypD dependent and an alternative route that is CypD independent; both pathways have been implicated in TBI. It is currently unclear on the amount of interplay between these two separate cell death pathways. There is growing evidence that suggests that CypD knockout is protective in neurodegenerative disorder models [72,73]. Interestingly, one study found that CypD-dependent cell death pathway is more critical in the progression of brain injury after hypoxic-ischemic insult [74]. This study further demonstrated that CypD knockout significantly reduced hypoxic-ischemic brain injury in adult mice while knockout exacerbated the injury in neonatal mice, highlighting age-dependent effects. Indeed, it has been shown that CypD levels in brain mitochondria increase with age, leading to dysfunction of F-ATP synthase [75]. Again, our results indicate that maintenance of mitochondrial homeostasis, by the absence of CypD, following TBI is protective.

We have previously demonstrated that continuous infusion of CsA for up to 7 days produces greater protection than a single dose administration of CsA following TBI [33]. Moreover, we have recently shown that the earlier the treatment with CsA is initiated following TBI the greater the neuroprotection that is afforded [35]. Taken together these results suggest that the mechanisms underlying cell death following TBI are an on-going process and that early intervention provides the greatest benefit. Past research demonstrated that NIM811 and CsA are equally as effective at providing neuroprotection in WT mice [47]. Here, we compared our own findings to these previously studies [33,35] by examining cortical tissue sparing also at 7 days post-injury. We show that CsA administration provides additive neuroprotection via non-CypD targeted biological pathway (Figure 5).

CypD knockout was shown to protect neurons against glutamate triggered cell death in vitro [55]. Importantly, the protection was dependent on the severity of the glutamate insult such that protection was not observed following the more severe insult. This may indicate that following more severe insults that CypD independent cell death mechanisms are more prevalent. In unpublished data from our lab, we showed that following a mild (0.5 mm) CCI TBI there is modest neuroprotection.

There are several limitations to this study. Only male mice were used to examine the effects of genetic knockout of CypD after TBI. Growing evidence suggests sex-specific responses to TBI [76]; therefore, future studies should examine sex in the context of CypD-dependent cell death after experimental brain injury. Further, breeding of CypD KO animals posed a challenge, which resulted in low n/group for comparison to WT. Of course, it would be interesting to investigate the role of CypD in other models of experimental TBI, such as a mild closed head injury model [60]. Finally, the only cognitive task utilized was the MWM. We recognize that other cognitive assays, such as novel object recognition, could be more sensitive to deficits and future studies should incorporate additional assays to fully examine the role of CypD in TBI-induced cognitive dysfunction.

Although knockout of CypD leads to significant neuroprotection, we observed that this does not lead to cognitive restoration. In fact, CypD KO animals display greater cognitive impairment. This could be explained by impaired calcium signaling in the absence of CypD or mPT as a necessary process to the recovery after TBI, whereby mitochondrial calcium scavenging and removal of dysfunction cells could result in overall neural network function. Nevertheless, our results suggest that CypD plays a prominent role in cell death mechanisms after TBI, as CypD ablation resulted in neuroprotection after TBI. This is the first study to report the effects of CypD knockout in a model of TBI and we showed that the lack of CypD significantly decreased neuronal damage associated with TBI. Thus, CypD-dependent cell death is critically involved in TBI, supporting the hypothesis that CypD is a valid target for neuroprotection following TBI.

## Figures and Tables

**Figure 1 cells-10-00199-f001:**
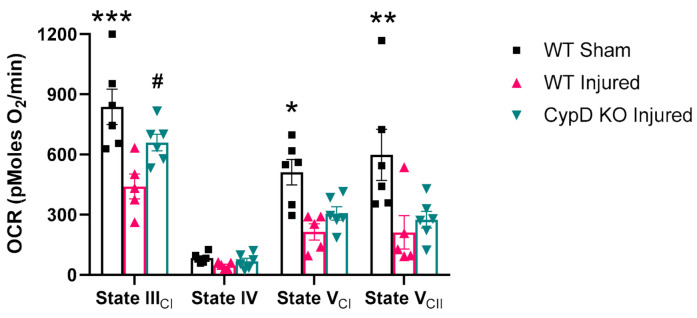
Cyclophilin D (CypD) knockout attenuates mitochondrial dysfunction following traumatic brain injury (TBI). At 6 h post-injury, complex I driven State III respiration was significantly reduced compared to sham in wild-type (WT) animals. However, CypD knockout attenuated State III mitochondrial dysfunction after TBI. For State V respiration, there was no significant difference between WT Injured and CypD KO Injured. * *p* = 0.0046 vs. WT Injured, ** *p* < 0.002 vs. WT Injured and knockout (KO) Injured, *** *p* = 0.0001 vs. WT Injured, ^#^
*p* = 0.047 vs. WT Injured. Five µg mitochondrial protein were added to each well. Data points represent group mean ± SEM. N = 5–6/group.

**Figure 2 cells-10-00199-f002:**
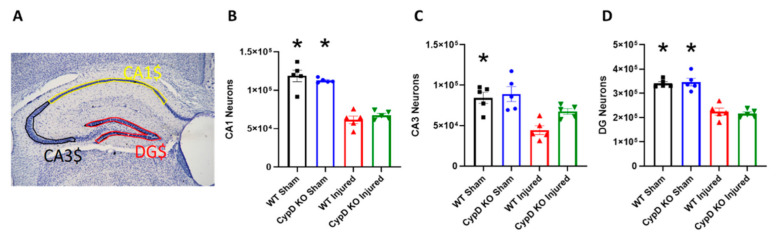
CypD knockout mitigates TBI-related CA3 neuronal loss. (**A**) Outline of hippocampal sub-regions, DG, CA3, and CA1. (**B**) Injury significantly decreased the number of CA1 neurons in both WT and CypD KO mice. (**C**) There was a significant decrease in CA3 neurons after TBI in WT mice. However, CypD knockout protected CA3 neurons from injury-induced cell loss. (**D**) TBI resulted in a significant decrease of DG neurons in both WT and CypD KO mice. * *p* < 0.003 compared to injured counterpart in each genotype. Data points represent group mean ± SEM. N = 5/group.

**Figure 3 cells-10-00199-f003:**
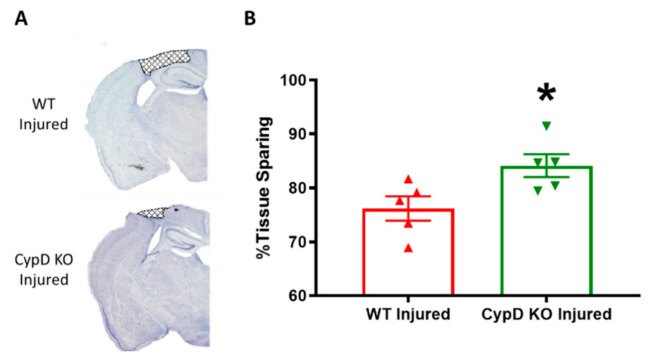
CypD knockout increases tissue sparing following TBI. (**A**) Representative coronal brain sections displaying lesion area (bregma level −1.4 mm). (**B**) Quantitative assessment of tissue sparing revealed that CypD knockout animals had significantly higher tissue sparing percentage compared to WT animals. * *p* = 0.033. Data points represent group mean ± SEM. N = 5/group.

**Figure 4 cells-10-00199-f004:**
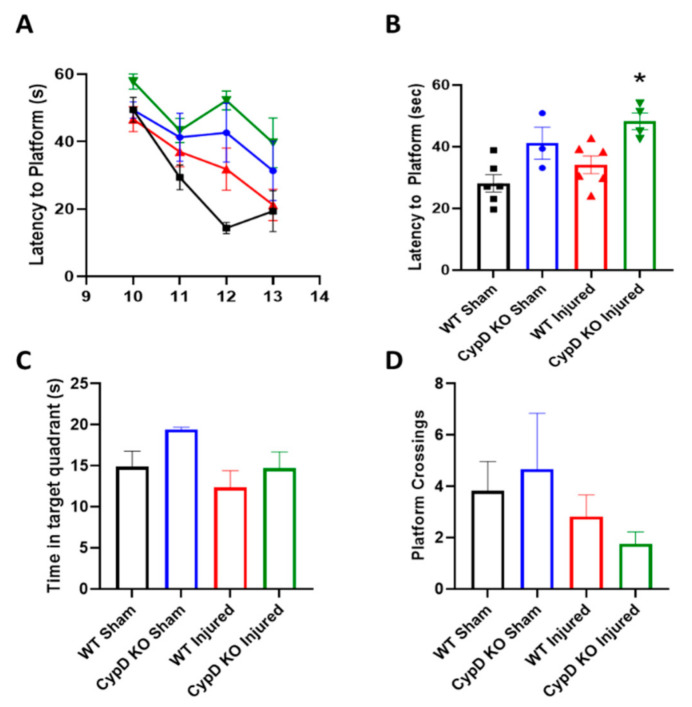
CypD knockout does not improve memory function following TBI. (**A**) Average learning progression, characterized by latency to platform, over days 10 to 13 after injury. (**B**) Morris water maze (MWM) latency to platform data collapsed across all training days. CypD knockout in injured animals resulted in an increase in latency to platform compared to WT Injured mice. * *p* = 0.033 vs. WT Injured. (**C**) At 14 d after TBI, mice performed the probe trial. Time in the target quadrant was calculated. (**D**) During the probe trial, the number of times mice crossed the platform area was also quantified. Data points represent group mean ± SEM. N = 3–6/group.

**Figure 5 cells-10-00199-f005:**
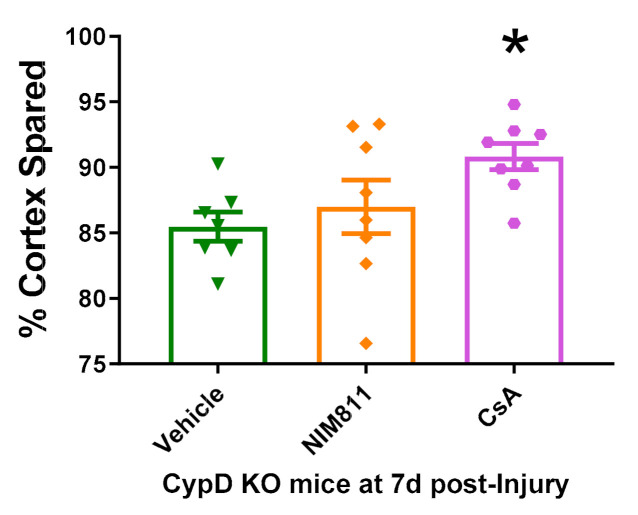
Cyclosporin A (CsA) and NIM811 treatment in CypD KO animals following TBI. While NIM811 did not alter tissue sparing in CypD KO animals compared to vehicle, CsA administration improved tissue sparing following experimental TBI mice. * *p* = 0.039 vs. Vehicle. Data points represent group mean ± SEM. N = 7–8/group.

## Data Availability

The data presented in this study are available on request from the corresponding author.

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
