# Peer review of "Genetic Approach to Elucidate the Role of Cyclophilin D in Traumatic Brain Injury Pathology"

_cells, 2021, doi:10.3390/cells10020199_

Round 1
Reviewer 1 Report
In this work, the authors evaluate the role of CypD in neuropathology post- TBI, performed on Knock out mice lacking the CypD encoding gene Ppif.
The findings were summarized as follows:
- knockout restored acute mitochondrial function following TBI.
- following injury, cortical tissue sparing and CA3 neuron density are improved in mice lacking the gene in the subacute phase
- No Cognitive benefit (MWM testing) was observed.
There are major concerns:
The endpoint that the authors chose are not sufficient and more experiments are needed:
First, neuronal survival and neuroprotection were based on cortical and hippocam[al neuronal count. Performing CCI should be coupled with counting neuronal cell death and evaluate how the CySA or the KO or NIM11 have decreased cell death. Hus, apoptotic markers should be evaluated
Second, no neuroinflammatory markers were assessed to see if microglial and astrocytic activation was affected by the genetic treatment and drug intervention.
Third, the behavioral assessment was based on MWM which evaluates memory and, surprisingly, such elegant experts in the field of TI involving Dr. Sullivan and Dr. Geddes have based their behavioral assessment only on MWM!!!! Besides using 5-8 animals is not acceptable in behavioral testing
Forth, mitochondrial bioenergetics should be evaluating the expression of different complexes and I would suggest performing oxo-phos antibody Western blotting that can evaluate different complexes at one time
Fifth, Oxidative stress markers such as catalase, SOD and catalase should be evaluated
Finally, tissue-sparing is considered non-accurate to say least in assessing neuroprotection as it is so subjective and the sections showed in the paper are taken from the different brain region cutting as it seems.
Minor Comments:
Please spell CDC:
Please correct spelling: the pore of the mPT
In the section of Therapeutic targeting of mPTP by Cyclosporin A, the Authors start discussing what is CYPD:
This should be moved to an earlier section
. CypD belongs to a family of proteins known as peptidyl-prolyl cis-trans isomerases (PPIases) and is localized in the mitochondrial matrix. This was elucidated by the observed effect of mPTP desensitization after CsA administration, taking higher levels of intra-mitochondrial calcium to initiate pore formation
Author Response
The authors would like to thank each reviewer for taking the time and effort to review and make suggestions to improve the quality of the paper. Please see our point-to-point responses in italics.
Reviewer 1
In this work, the authors evaluate the role of CypD in neuropathology post- TBI, performed on Knock out mice lacking the CypD encoding gene Ppif. The findings were summarized as follows, 1) knockout restored acute mitochondrial function following TBI, 2) following injury, cortical tissue sparing and CA3 neuron density are improved in mice lacking the gene in the subacute phase, 3) no Cognitive benefit (MWM testing) was observed.
There are major concerns:
The endpoint that the authors chose are not sufficient and more experiments are needed:
First, neuronal survival and neuroprotection were based on cortical and hippocampal neuronal count. Performing CCI should be coupled with counting neuronal cell death and evaluate how the CySA or the KO or NIM11 have decreased cell death. Hus, apoptotic markers should be evaluated.
We feel that our unbiased approach for neuronal counting and tissue sparing assessment is more than adequate to assess neuroprotection and neuronal death. With Nissl staining, neurons are distinct and, in most cases, it is easy to distinguish between neurons and glia based on nuclear and cell body staining [1]. Consideration of cell shape and size in neuron counting parameters has been added to the Methods sections (line 268). While apoptosis is an important cell death mechanism, the mode of cell death was not a focus of this study.
Second, no neuroinflammatory markers were assessed to see if microglial and astrocytic activation was affected by the genetic treatment and drug intervention.
While we agree that neuroinflammation is important after TBI, the focus of the current manuscript is the effect of CypD on mitochondrial function. Microglial/astrocytic analysis is beyond the scope of this manuscript.
Third, the behavioral assessment was based on MWM which evaluates memory and, surprisingly, such elegant experts in the field of TI involving Dr. Sullivan and Dr. Geddes have based their behavioral assessment only on MWM!!!! Besides using 5-8 animals is not acceptable in behavioral testing
A common failing of preclinical research is conducting too many behavioral assays with a cohort of animals, without days of rest. Considering animals were only alive for 18 days after injury and the MWM assay is a five-day task, this was the only behavior test we could perform to ensure a week of recovery after injury (line 180). Of course, other behavioral tasks, such as novel object recognition, measure different aspects of cognition and have varying sensitivity. We acknowledge this as a limitation in the Discussion (line 459-461). In response to the n/group, based on power analysis, we could observe significant differences based on expected mean and variance (see this addition to the Statistics section; line 280-284).
Forth, mitochondrial bioenergetics should be evaluating the expression of different complexes and I would suggest performing oxo-phos antibody Western blotting that can evaluate different complexes at one time
While we agree that this assessment could be instructive to assess if recovery is functional or structural, this would require the generation of another cohort of animals. Due to a number of factors, including COVID-related shutdowns, the mouse colony has ceased to exist so generation of additional animals is not possible.
Fifth, Oxidative stress markers such as catalase, SOD and catalase should be evaluated
Indeed, oxidative stress and mitochondrial function are closely related processes and oxidative stress is important after TBI. However, we are not able to perform additional studies based on the factors at our institution. The effects that CypD knockout could have on mitochondrial-related oxidative damage based are far-reaching and would be speculative at this point, therefore commenting on this is beyond the scope of the current paper.
Finally, tissue-sparing is considered non-accurate to say least in assessing neuroprotection as it is so subjective and the sections showed in the paper are taken from the different brain region cutting as it seems.
This technique is an unbiased approach that is routinely used in our laboratory and many others and utilizes the Cavalieri method of segmentation to compare ipsilateral cortex to contralateral cortex. This analysis has been performed by blinded operators as previously specified in the methods section and the methodology is detailed in the methods (lines 251- 261). The tissue sparing sections used as representative images were both taken at bregma level -1.4 mm; however, CCI occasionally distorts hippocampal morphology as in the case of the representative CypD KO image.
Minor Comments:
Please spell CDC:
Thank you, this has been changed (line 38)
Please correct spelling: the pore of the mPT
This has been clarified in the text (line 49, 69 for example).
In the section of Therapeutic targeting of mPTP by Cyclosporin A, the Authors start discussing what is CYPD:
This should be moved to an earlier section
“CypD belongs to a family of proteins known as peptidyl-prolyl cis-trans isomerases (PPIases) and is localized in the mitochondrial matrix. This was elucidated by the observed effect of mPTP desensitization after CsA administration, taking higher levels of intra-mitochondrial calcium to initiate pore formation”
While we agree it is important to discuss CypD earlier in the manuscript, considering the style of the mini-review, we feel where this is located is best for manuscript flow.
- Zhu, Y., et al., Comparison of unbiased estimation of neuronal number in the rat hippocampus with different staining methods. J Neurosci Methods, 2015. 254: p. 73-9.
Reviewer 2 Report
The manuscript by Readnower et al. describe a series of interesting experiments clarifying the role of mitochondrial cyclophilin D (CypD) and mitochondrial permeability transition pore (mPTP) in traumatic brain injury (TBI) pathology. The authors are experts in TBI and mitochondrial research. In this study, the authors used immunosuppressant cyclosporin A (CsA) and its non-immunosuppressive analog NIM811 applied to brain tissue shortly after TBI. In addition, the authors used CypD-knockout mice to investigate the role of CypD in the secondary brain injury following mechanical brain trauma. The authors provide compelling evidence that CypD deletion provides neuroprotection following TBI. Surprisingly, the authors found additional neuroprotection against TBI by applying CsA to CypD-KO brain tissue. This was interpreted as evidence for additional CypD-independent CsA targets contributing to neuroprotection. The manuscript deals with important issue of neuroprotection following TBI, reports very intriguing findings, and easy to read. There is no doubt that this study will be of great interest to neuroscientist, cell biologists, and biochemists interested in the role of mPTP in TBI pathology. There are just a few issues, which require the authors’ attention.
Line 39: It would be helpful to indicate the time period, in which 3 billion TBI-related emergency visits took place.
Line 51: "The mPTP...allows the outflow of...cytochrome c..." - This statement seems not quite accurate. An induction of the PTP per se does not allow the outflow of cytochrome c, but PTP induction leads to mitochondrial swelling and rupture of the outer mitochondrial membrane, causing the release of cytochrome c.
Line 62: "...recent evidence suggests that while these 62 mitochondrial inner membrane proteins contribute to mitochondrial desensitization to high calcium 63 levels, they may not constitute the pore of the mPT [6,7]." - There are several recent studies that cast doubt on the role of ATP synthase in mPTP. Citing these papers may help to better balance the authors' view on the role of different proteins in PTP:
Carroll J, He J, Ding S, Fearnley IM, Walker JE (2019) Persistence of the permeability transition pore in human mitochondria devoid of an assembled ATP synthase. Proc Natl Acad Sci U S A 116:12816-12821.
He J, Carroll J, Ding S, Fearnley IM, Walker JE (2017a) Permeability transition in human mitochondria persists in the absence of peripheral stalk subunits of ATP synthase. Proc Natl Acad Sci U S A 114:9086-9091.
He J, Ford HC, Carroll J, Ding S, Fearnley IM, Walker JE (2017b) Persistence of the mitochondrial permeability transition in the absence of subunit c of human ATP synthase. Proc Natl Acad Sci U S A 114:3409-3414.
Zhou W, Marinelli F, Nief C, Faraldo-Gomez JD (2017) Atomistic simulations indicate the c-subunit ring of the F1Fo ATP synthase is not the mitochondrial permeability transition pore. Elife 6.
In addition, there are at least two recent papers, which demonstrate that the ANT can form a large ion channel and, thus, can contribute to mitochondrial permeability transition phenomenon:
Neginskaya MA, Solesio ME, Berezhnaya EV, Amodeo GF, Mnatsakanyan N, Jonas EA, Pavlov EV (2019) ATP Synthase C-Subunit-Deficient Mitochondria Have a Small Cyclosporine A-Sensitive Channel, but Lack the Permeability Transition Pore. Cell Rep 26:11-17.
Karch J, Bround MJ, Khalil H, Sargent MA, Latchman N, Terada N, Peixoto PM, Molkentin JD (2019) Inhibition of mitochondrial permeability transition by deletion of the ANT family and CypD. Sci Adv 5:eaaw4597.
Line 75: "CypD, a target of the FDA approved immunosuppressant cyclosporin A (CsA), has been shown to play a key role in the modulation of the mitochondrial permeability transition pore (mPTP) [1].” - Please, double check this Reference. I doubt that Ref. 1 reports that CypD plays a key role in the modulation of the mitochondrial permeability transition pore.
Line 117: Here is a second sentence in a row starting with "Additionally". Sounds awkward.
Line 126: "Studies have demonstrated the role CypD plays in calcium uptake and mPTP formation in mitochondria [42,43]." - In fact, CypD does not play a role in calcium uptake, but it plays a role in mPTP formation, that limits calcium uptake by mitochondria.
Line 140: "CypD-mediated 140 sensitization to mitochondrial calcium stress is also observed in brain mitochondria, where knockout of CypD improved mitochondrial stability in the presence of excess calcium [44].” - There is another, earlier study with isolated brain mitochondria and neurons, indicating CypD involvement in mitochondrial sensitivity to elevated calcium:
Li, V.; Brustovetsky, T.; Brustovetsky, N. Role of cyclophilin D-dependent mitochondrial permeability transition in glutamate-induced calcium deregulation and excitotoxic neuronal death. Experimental neurology 2009, 218, 171-182, doi:10.1016/j.expneurol.2009.
Line 281: It would be very helpful for readers who are not familiar with mitochondrial research to disclose the meaning of States of mitochondrial respiration (State III, State V, etc.)
Author Response
The authors would like to thank each reviewer for taking the time and effort to review and make suggestions to improve the quality of the paper. Please see our point-to-point responses in italics.
Reviewer 2
The manuscript by Readnower et al. describe a series of interesting experiments clarifying the role of mitochondrial cyclophilin D (CypD) and mitochondrial permeability transition pore (mPTP) in traumatic brain injury (TBI) pathology. The authors are experts in TBI and mitochondrial research. In this study, the authors used immunosuppressant cyclosporin A (CsA) and its non-immunosuppressive analog NIM811 applied to brain tissue shortly after TBI. In addition, the authors used CypD-knockout mice to investigate the role of CypD in the secondary brain injury following mechanical brain trauma. The authors provide compelling evidence that CypD deletion provides neuroprotection following TBI. Surprisingly, the authors found additional neuroprotection against TBI by applying CsA to CypD-KO brain tissue. This was interpreted as evidence for additional CypD-independent CsA targets contributing to neuroprotection. The manuscript deals with important issue of neuroprotection following TBI, reports very intriguing findings, and easy to read. There is no doubt that this study will be of great interest to neuroscientist, cell biologists, and biochemists interested in the role of mPTP in TBI pathology. There are just a few issues, which require the authors’ attention.
We genuinely thank the reviewer for the kind comments and remarks regarding the impact of this manuscript.
Line 39: It would be helpful to indicate the time period, in which 3 billion TBI-related emergency visits took place.
Thank you, there was a typing error in this original figure, which has been corrected to “3 million… yearly”; this has been amended (line 40).
Line 51: "The mPTP...allows the outflow of...cytochrome c..." - This statement seems not quite accurate. An induction of the PTP per se does not allow the outflow of cytochrome c, but PTP induction leads to mitochondrial swelling and rupture of the outer mitochondrial membrane, causing the release of cytochrome c.
Thank you for the close attention to this statement. We wholeheartedly agree and this has now been revised (line 52-53).
Line 62: "...recent evidence suggests that while these 62 mitochondrial inner membrane proteins contribute to mitochondrial desensitization to high calcium 63 levels, they may not constitute the pore of the mPT [6,7]." - There are several recent studies that cast doubt on the role of ATP synthase in mPTP. Citing these papers may help to better balance the authors' view on the role of different proteins in PTP:
-Carroll J, He J, Ding S, Fearnley IM, Walker JE (2019) Persistence of the permeability transition pore in human mitochondria devoid of an assembled ATP synthase. Proc Natl Acad Sci U S A 116:12816-12821.
-He J, Carroll J, Ding S, Fearnley IM, Walker JE (2017a) Permeability transition in human mitochondria persists in the absence of peripheral stalk subunits of ATP synthase. Proc Natl Acad Sci U S A 114:9086-9091.
-He J, Ford HC, Carroll J, Ding S, Fearnley IM, Walker JE (2017b) Persistence of the mitochondrial permeability transition in the absence of subunit c of human ATP synthase. Proc Natl Acad Sci U S A 114:3409-3414.
-Zhou W, Marinelli F, Nief C, Faraldo-Gomez JD (2017) Atomistic simulations indicate the c-subunit ring of the F1Fo ATP synthase is not the mitochondrial permeability transition pore. Elife 6.
We thank you for illuminating these papers. We now cite these as well to have a well-balanced take of the mechanism of the mPTP (line 73).
In addition, there are at least two recent papers, which demonstrate that the ANT can form a large ion channel and, thus, can contribute to mitochondrial permeability transition phenomenon:
-Neginskaya MA, Solesio ME, Berezhnaya EV, Amodeo GF, Mnatsakanyan N, Jonas EA, Pavlov EV (2019) ATP Synthase C-Subunit-Deficient Mitochondria Have a Small Cyclosporine A-Sensitive Channel, but Lack the Permeability Transition Pore. Cell Rep 26:11-17.
-Karch J, Bround MJ, Khalil H, Sargent MA, Latchman N, Terada N, Peixoto PM, Molkentin JD (2019) Inhibition of mitochondrial permeability transition by deletion of the ANT family and CypD. Sci Adv 5:eaaw4597.
Thank you. These have been added as well (line 66).
Line 75: "CypD, a target of the FDA approved immunosuppressant cyclosporin A (CsA), has been shown to play a key role in the modulation of the mitochondrial permeability transition pore (mPTP) [1].” - Please, double check this Reference. I doubt that Ref. 1 reports that CypD plays a key role in the modulation of the mitochondrial permeability transition pore.
Thank you for catching this oversight. This has now been updated (line 83).
Line 117: Here is a second sentence in a row starting with "Additionally". Sounds awkward.
This has been revised now (line 124).
Line 126: "Studies have demonstrated the role CypD plays in calcium uptake and mPTP formation in mitochondria [42,43]." - In fact, CypD does not play a role in calcium uptake, but it plays a role in mPTP formation, that limits calcium uptake by mitochondria.
Thank for highlighting this distinction – we have now re-phrased this statement to make it accurate (line 134).
Line 140: "CypD-mediated 140 sensitization to mitochondrial calcium stress is also observed in brain mitochondria, where knockout of CypD improved mitochondrial stability in the presence of excess calcium [44].” - There is another, earlier study with isolated brain mitochondria and neurons, indicating CypD involvement in mitochondrial sensitivity to elevated calcium:
Li, V.; Brustovetsky, T.; Brustovetsky, N. Role of cyclophilin D-dependent mitochondrial permeability transition in glutamate-induced calcium deregulation and excitotoxic neuronal death. Experimental neurology 2009, 218, 171-182, doi:10.1016/j.expneurol.2009.
Thank you for drawing our attention to this – we now include this reference (line 151).
Line 281: It would be very helpful for readers who are not familiar with mitochondrial research to disclose the meaning of States of mitochondrial respiration (State III, State V, etc.)
We now define these states (line 297-301).
Reviewer 3 Report
In this study the authors examined the role of cyclophilin D (CypD) in mechanisms of traumatic brain injury (TBI). To this end CypD knockout mice (CypD KO) and CypD inhibitors CsA, and NIM811 were used in TBI experiments. This is carefully conducted study, although with limited number of animals. Furthermore, there are couple of points that need to be addressed.
- The Introduction is too long. It should be shortened.
- Page4, Line 142: it is not clear what the authors mean by “mitochondrial stability”.
- Reference related to mitochondrial respiratory function measurements by seahorse are not listed in the References list.
- The amount of mitochondrial protein used for Seahorse experiments is not stated. Furthermore, the respiratory rates are expressed in pMol O2/min, it should be pMol O2/min/mg protein.
- The authors conclude that the absence of CypD is neuroprotective since the number of surviving CA3 neurons in TBI injured CypD KO animals is not significantly different from the number of CA3 neurons in sham CypD KO mice. However, the number of CA3 surviving neurons in injured CypD KO mice is also not significantly higher when compared to number of CA3 neurons in WT TBI animals. Thus, the CypD KO does not show improvement comparing to WT animals.
- The effect of CypD on cortical tissue sparing in CypD KO mice was evaluated at 18 days following TBI. However, the effect of CypD inhibitors (CsA and NIM811) was determined at 7 days of recovery. Whys this discrepancy?
- Page 10, line 401-402: authors need to mention and discuss the study showing age depend expression levels of CypD in brain tissue.
- Similarly, the differential expression of CypD in different brain subregions and cell-types should be discussed.
Author Response
The authors would like to thank each reviewer for taking the time and effort to review and make suggestions to improve the quality of the paper. Please see our point-to-point responses in italics.
Reviewer 3
In this study the authors examined the role of cyclophilin D (CypD) in mechanisms of traumatic brain injury (TBI). To this end CypD knockout mice (CypD KO) and CypD inhibitors CsA, and NIM811 were used in TBI experiments. This is carefully conducted study, although with limited number of animals. Furthermore, there are couple of points that need to be addressed.
Thank you kindly for the review and feedback.
- The Introduction is too long. It should be shortened.
The editor welcomed a manuscript with a mini-review, which is why the Introduction is comparatively lengthy.
Page4, Line 142: it is not clear what the authors mean by “mitochondrial stability”.
We have now re-worded this to “mitochondrial membrane polarization and survival” to increase clarity (line 149-150).
Reference related to mitochondrial respiratory function measurements by seahorse are not listed in the References list.
We have now updated the references for this (line 222).
The amount of mitochondrial protein used for Seahorse experiments is not stated. Furthermore, the respiratory rates are expressed in pMol O2/min, it should be pMol O2/min/mg protein.
Indeed, we apologize for the oversight and now state that 5 µg of protein were added to each well (line 222-223; 312), as such the raw O2/min values are comparable across all wells.
The authors conclude that the absence of CypD is neuroprotective since the number of surviving CA3 neurons in TBI injured CypD KO animals is not significantly different from the number of CA3 neurons in sham CypD KO mice. However, the number of CA3 surviving neurons in injured CypD KO mice is also not significantly higher when compared to number of CA3 neurons in WT TBI animals. Thus, the CypD KO does not show improvement comparing to WT animals.
Thank you for the distinction. We now re-phrase our interpretation of the data (line 314; 323; 325).
The effect of CypD on cortical tissue sparing in CypD KO mice was evaluated at 18 days following TBI. However, the effect of CypD inhibitors (CsA and NIM811) was determined at 7 days of recovery. Whys this discrepancy?
This is a valid point. The main reason was due to the lack of behavioral test performed in the CypD inhibitor study and 7 days post-injury being used in our previous studies with CsA in TBI (see Sullivan, et al. 2000 Exp Neurology; Sullivan, et al. 2011 J Neurotrauma)[1, 2]. This has been commented on is the Discussion (line 393-395; 440-441).
Page 10, line 401-402: authors need to mention and discuss the study showing age depend expression levels of CypD in brain tissue.
We have added Gauba, et al. 2017 JAD to this discussion (line 430-431).
Similarly, the differential expression of CypD in different brain subregions and cell-types should be discussed.
We have added and now discussed Hazelton, et al. 2009 (line 406-409).
REFERENCES
- Sullivan, P.G., M. Thompson, and S.W. Scheff, Continuous infusion of cyclosporin A postinjury significantly ameliorates cortical damage following traumatic brain injury. Exp Neurol, 2000. 161(2): p. 631-7.
- Sullivan, P.G., A.H. Sebastian, and E.D. Hall, Therapeutic window analysis of the neuroprotective effects of cyclosporine A after traumatic brain injury. J Neurotrauma, 2011. 28(2): p. 311-8.
Round 2
Reviewer 1 Report
We thank the authors for the response, while I agree with the Covid-related reasons, I think the neuroinflammation work should be performed in this context of the proposed research. Performing Iba1 and GFAP staining or Wb can be easily achieved.
Reviewer 3 Report
The manuscript is improved. All reviewer comments were addressed.